# Ecological Sanitation and Sustainable Nutrient Recovery Education: Considering the Three Fixes for Environmental Problem-Solving

**Julian Junghanns** [1,*] and **Thomas Beery** [2]

1    Faculty of Natural Sciences, Kristianstad University, SE-291 88 Kristianstad, Sweden
2    Faculty of Education, Kristianstad University, SE-291 88 Kristianstad, Sweden; thomas.beery@hkr.se
*    Correspondence: jt.junghanns@gmx.de

**Abstract:** In the context of phosphorus as a finite resource and the unsustainable character of current sanitation in Europe, this paper examined social factors in a technological transition towards sustainable sanitation. The evaluation is based on the idea of cognitive, structural, and technological fixes to achieve environmental protection. The cognitive fix has been evaluated through literature and a European-wide survey with universities that offer civil and environmental engineering programs. Contrary to an initial hypothesis, ecological sanitation and nutrient recycling are taught by the majority (66%) of responding programs. There are, however, local differences in terms of context and detail of the education. The main impediments for teaching were identified as academic resources (especially in Belgium, Germany and Denmark) and the technological status quo (Ireland, Italy, Spain and some programs of the United Kingdom). Instructors' personal commitment and experience was evaluated to be a key factor for an extensive coverage of sustainable sanitation in higher education programs. The role of higher education has a critical role to play in changing sanitation practices, given the unique professional developmental stage of students and the potential for a cognitive fix to contribute to meaningful change.

**Keywords:** cognitive fix; nutrient recycling; education; human fertilizer; disgust; attitudes; phosphorus; wastewater; sustainable sanitation; status quo.

## 1. Introduction

The need for sanitation to adapt to the principles of sustainability has become more and more evident in recent years. For example, there have been global targets to meet sustainable development goals like the 8 Millennium Development Goals (MDGs) set in the year 2000 [1]. A more significant focus on sustainable sanitation, however, has been established with the introduction of the 17 Sustainable Development Goals (SDGs) in 2015 by the United Nations. The SDGs have been adopted by all member states of the United Nations [1], showing a significant commitment to sustainability on a global scale. Among the goals, SDG 6 sets a clear focus on clean water supply and sanitation. Goal 6 target 6.3 states that an improvement of water quality should be achieved by "reducing pollution, eliminating dumping and minimizing release of hazardous chemicals and materials, halving the proportion of untreated wastewater and substantially increasing recycling and safe reuse globally" [1] (p. 20) by 2030 [1,2]. This goal and target provide the context for the study presented in this paper. And despite concern that the SDGs may lead to further promotion of end of the pipe or technology-only solutions rather than the development of sustainable systems [3], we posit that sustainable sanitation and nutrient recycling provide a path forward in accordance with these global goals.

## 1.1. Sustainable and Ecological Sanitation

Sustainable sanitation (SS) is defined as a system that is "economically viable, socially acceptable, technically and institutionally appropriate, and [that] protect[s] the environment and natural resources" [4]. It's a system that considers all dimensions of sustainability (criteria) [4]. At the moment, only a few of the sanitation systems in Europe are considered sustainable [5–7]. The majority of conventional sanitation systems present linear end-of-pipe systems that serve human health and the environment by disposing of waste and wastewater by several different treatment steps. The recycling and reuse of wastewater and dissolved material are often not considered in the wastewater treatment process [8]. An SS system, however, should focus on using the ecosystem services provided by human wastewater, such as the recycling of nutrients (phosphorus, nitrogen, and potassium), organic matter, and water or the production of excreta-derived biogas (energy) [4]. In addition, rare-earth elements and valuable metals could be recovered [9]. A recent study described 17 potential ecosystem services within sanitation, including nutrient recycling, among others [10].

Of the nutrients available in wastewater, phosphorus (P) is an indispensable substance in agricultural fertilizers with a finite global supply and an uneven distribution around the world. Phosphorus is increasingly contaminated with heavy metals and radioactives [11–16]. Given such challenges, its recycling from wastewater is thus of great importance. Potential recycling products include calcium phosphate ($Ca_3(PO_4)_2$), struvite (MAP, $MgNH_4PO_4 \cdot 6H_2O$), or magnesium potassium phosphate hexahydrate (MPP, $MgKPO_4 \cdot 6H_2O$). MAP adds nitrogen (N) to the list of recycled substances, while MPP adds the nutrient potassium (K). To produce MPP, one needs to extract all available ammonium first, while for the production of MAP this is a key substance. Thus, MAP and MPP cancel each other out [5,17]. By taking a circular economy (CE) perspective, human wastewater and the dissolved P could be considered a resource rather than waste; these and other resources of municipal wastewater could be managed for human reuse [6,7,9,13,18,19]. CE has been criticized for its lack of clear definition [20] and often missing link to the social sphere of sustainability [21,22], but could be an indispensable tool to achieve many of the targets of the SDGs, among them the recycling of human wastewater (6.3) [21–23]. Human wastewater could, for example, be transformed into human fertilizer for the agricultural sector as it is conceptualized in *Ecological Sanitation* (EcoSan) [10,24,25]. EcoSan is characterized by decentral (household management) or semi-decentral (communal management) sanitation systems that have a strong focus on nutrient recovery through the collection of urine and feces. Among other collection techniques, source separation is recommended [26,27]. Urine and feces can be reused as fertilizer in the agricultural sector, especially because close to 100% of P consumed through food is also excreted [12]. Especially urine is recommended to be transformed into fertilizer because its proportionate share of NPK is significant [28]. In relation to factors such as diet, age, and geographical location, urine offers approximately 90% of N (initially in the form of urea [28]), 60 to 76% of P, and 50 to 80% of K available in excreta [25,29].

There are only a few European implementations of ecological sanitation mentioned in the literature [19,27], and this may be due to its decentralized character; EcoSan is mostly applied in developing countries. In contrast, in developed countries, an adaption of the conventional centralized sanitation system towards sustainability is more frequently used [30]. While different geographical implementations are followed by either a central or decentral system, one needs to emphasize that this is not the only difference between applications in developing and industrialized countries. Sanitation systems that aim for sustainability might use different nutrient recycling technologies than EcoSan systems. In an SS system, these technologies have to incorporate sustainability dimensions and have to be economically and socially viable. EcoSan is a resource-oriented sanitation (ROS) system that might be sustainable but does not have to be. In fact, EcoSan projects in Burkina Faso were evaluated to be unsustainable in the long run [31].

Research would suggest that an effective transition to EcoSan or SS would require a combination of technological, infrastructural, and educational efforts. Heberlein [32] refers to these as technological, (infra-)structural, and cognitive fixes. A "fix" refers to the environmental problem-solving efforts,

techniques, or strategies, and Heberlein [32] highlights the contributions of these different domains of problem-solving. For example, there is extensive research suggesting technologies to ease the transition to sustainable sanitation systems [19,33]. However, lacking is research into infrastructural and educational approaches. In addition, the social factors involved in such a transition are less vigorously explored and need more in-depth consideration [9]. Thus, this article focuses on the human and societal factors of ecological sanitation and nutrient recycling. The main focus will be set on the potential role of a cognitive fix to help address this environmental challenge [32]. The cognitive fix is defined as information and knowledge to guide understanding and support behavior change [34].

Universities may be instrumental in regards to this cognitive fix. Universities and their faculties play an essential role in societal changes [35], and by taking a more participatory approach, their students can become future change agents [36,37]. When understanding academic education as a chance to enable a permanent attitude change towards sustainable sanitation, one needs to question the knowledge level within the critical programs [38]. Herein, the research question is situated: is there a lack of knowledge about ecological sanitation and nutrient recycling technologies within the educational programs of future wastewater managers and sanitation engineers? And further, how important are these technologies to the university educators responsible for these wastewater programs?

### 1.2. Considering the Human Element

As previously mentioned, only a small fraction of all European sanitation systems are considered sustainable. Furthermore, there are only a few implementations of EcoSan in Europe, but the potential is much greater. So why is there such a big gap? Are we facing a status quo in the way we handle our sanitation? A status quo describes the current state of things but is perceived as a slightly negative term because of its resilient character. People want to keep the status quo (the "old" system) and are thus reluctant to change or progress [39]. The human element is identified as the main impediment when implementing technologies to recover resources [40]. A lack of needed knowledge, experience transfer (through excursions to lighthouse projects for example [41]), and commitment of the stakeholders involved in projects have been evaluated as crucial reasons for project failure [26,27]. Looking into and understanding social sciences and behavioral theory is thus an essential step in the transition towards a sustainable system in general [42–44] and SS in particular [9,14].

Coming from an anthropologic perspective, the roots of human behavior in regards to waste management may be an adaptive system for disease avoidance behavior [45]. Underlying this adaptive system may be the human emotional response of disgust, argued to be an evolutionary trait that helped humankind to protect itself from pathogens and which still influences our hygienic behavior [46]. Disgust is said to be a moral emotion that is deeply rooted within humans. It sometimes appears to be an innate response; other times, it emerges through social or group learning [47]. As a result of infection events like the great stink of London in 1858, feces and dirty water is commonly regarded as disgusting, and the contact with it has to be avoided at all costs [13,48]. Such disease avoidance behaviors are highly intuitive, automatic, and independent from new scientific knowledge [47]. A new concept like EcoSan that promotes the diversion of urine and feces, direct contact and management of these substreams may not be taken into consideration because disgust dictates the perception of these substreams. Scientific knowledge might indicate that this new process is a more sustainable way to deal with wastewater, but the emotion of disgust may prevent consideration of its implementation. Disgust might thus be one factor of many, and it needs consideration where it seemingly impedes (technological) progress.

It needs to be emphasized that the impact and strength of disgust is geographically and historically dependent. While wastewater and its treatment are commonly perceived negatively by the public in Europe, countries such as China do not report similar feelings and are pushing for wastewater reuse technologies [37,49]. Despite contemporary perceptions, the reuse of wastewater for agricultural irrigation, among other applications, has a long tradition in Europe and around the world. It was a common measure in Ancient Greek and Italian cities in the 14th century, as well as in the cities of

London and Norwich; further, it was the focus of political systems such as the national socialist party in Germany in the 1930s [48]. The development of the secondary treatment has been delayed due to World War II, which left many European countries with simple sanitation until 1950 [48]. During this time (1930–1940), many Swedish people were collecting their wastewater for garden irrigation [50]. These examples remind us that overcoming disgust and focusing on potential benefits may be possible in regards to (alternative) wastewater management applications.

The Three Fixes for Environmental Problem-Solving

When considering human behavior and environmental behavior change, an important source of guidance is research within the field of environmental education. Some of the early models of environmental behavior change emphasize a cognitive fix—for example, linear models based on a progression of knowledge leading to awareness and concern, and ultimately pro-environmental behavior [51]. Ongoing research and practice have shown these models to be too simplistic; subsequently, increasingly complex considerations of environmental behavior have emerged [51]. It is important to note that while an exclusive cognitive approach (the "cognitive fix") for understanding human behavior is inadequate, it does not mean that information and knowledge are not part of more complex descriptions of environmental behavior. For example, action-oriented and system thinking knowledge may be able to play an important role in environmental behavior change [52]. Recent work in environmental education reminds us that despite the lingering impact of the knowledge to behavior paradigm, we must strive to consider the multitude of factors that may contribute to environmental behavior change [52]. Given calls for thoughtful complexity, this current research promotes the idea that solving the human behavioral element of the question of wastewater treatment will also need a multifaceted approach such as the previously noted three fixes for environmental problem-solving [32]. The fixes include the technological fix, i.e., the idea that introducing a new technology may be able to circumvent human behavior, for example, renewable energy technologies that make green energy more accessible [53]. The cognitive fix for environmental problems involves the use of education to change people's attitudes and behavior; as noted, environmental education has been based on providing enough information to build knowledge to help people make decisions and change behaviors [34]. The structural fix attempts to change the situation people find themselves in by introducing norms, laws, and even infrastructure to change human behavior [32]; structural fixes may work by nudging new behaviors [54]. Nudging, in this context, refers to guiding the public into behaviors that they might otherwise not engage in [55]. There are several different examples of structural fixes or nudges. Using information about price developments, and offering tips how to conserve water within a household, could encourage residents to decrease their usage of water [56].

In Germany and Switzerland, there are a couple of relevant examples of structural fixes available when considering ROS. In Germany, the federal and provincial governments revised their sewage sludge regulation (AbfKlärV) in 2017. The recycling of P is not dependent on the quality of the input sludge, but rather the size of the wastewater treatment plant (WWTP) in question. Over the next 12 to 15 years, all WWTPs that serve 50,000 and more inhabitants will be required to implement a P recycling facility; WWTPs that serve fewer people are not included in the process [57,58]. Aside from these developments, Switzerland is another positive example of advancing towards the recycling of nutrients. In 2013, the federal council of Switzerland agreed on a Green Economy action program with 27 measures to implement a green economy and to transit Switzerland towards a resource-efficient society [59]. They also established the Ordinance on the Avoidance and Disposal of Waste (VVEA) to recycle P from sludge and animal and bone meal in 2016 [60]. In Zurich, the project "Phosphorus recovery from sewage sludge ash" (Phos4life) has been under development to test P-recovery products from incinerated sludge; this project provides an example of the potential overlap between structural and technological fixes, i.e., how new technology may be able to guide infrastructural process development. In Zurich, they found that by producing high-quality phosphoric acid from sludge ash, a recycling

efficiency of 95% could be achieved. In addition, other substances like metals, iron chlorides, and mineral residues can be recycled as well [61].

The definitions of the three fixes include several terms that have a distinct meaning specific to the field of environmental psychology: attitudes, behavior, and norms. An attitude can be the underlying principle or opinion of a behavior (action); both terms can be closely related, but they do not have to be [32]. Ajzen [62] evaluated that attitudes and behavior are related, but the specific nature of this relation is still unknown. Broad attitudes may have an indirect influence on a specific behavior, but Ajzen [62] theorized that other factors such as the perceived behavioral control and intention need further consideration. For example, research has shown that attitudes are often a component of action, but likely not solely sufficient in aims toward supporting environmental behavior or action [63]. More recently, Marcinkowski and Reid [64] considered the attitude to behavior relationship (A-B) by considering reviews of studies on the relationship in environmental education and other fields. They found statistically significant A-B relationships highlighted in previous research, but many with relatively moderate strength only. Marcinkowski and Reid [64] also found that the relationship varies with different attitudinal objects or referents. Relatedly, they claim that moderators always matter. If these results are applied to a consideration of ROS, the consistently negative public attitude towards fecal matter and sanitation is a potential problem in general [7,32]. However, perhaps the A-B relationship may be different for students studying sanitation? According to Heberlein [32], the cognitive fix may be most successful with self-selecting groups and with college students. Students are still developing their attitudes during their time at the university and could thus be successfully influenced in their understanding and opinion towards a topic. Therefore, educators from the field of sanitation could enable future implementation of (sustainable) nutrient recovery technologies and principles by inspiring their students on the subject. Of interest to this study is whether the role of "student" may be one of the key moderators that may be able to strengthen the A-B relationship in this case. With this consideration comes the acceptance that it may be that public attitudes are not the critical concern in regards to ROS and SS, but rather the professional attitudes of professional sanitation engineers.

Aside from attitudes, norms influence behavior. They can regulate what actions are taken and which are not. Norms are feelings of strong obligations to engage in prosocial behavior of an individual or a group [65]. One's moral norm influences actual behavior [66,67]. Norms come with (formal and informal) sanctions that provide additional incentives to act a certain way. Such sanctions are some level of social or societal punishment for acting against a norm [32]. Being held accountable to laws is an example of a formal norm, while being left alone at a party when talking about an unpopular topic, like sanitation, could be characterized as an informal norm [7,48,68]. A civil engineer, on the other hand, might be accepted and expected to talk about wastewater while the public avoids the topic [32]. Thus, one needs to further differentiate between public norms and norms established by societal and professional roles [69]. Comparing public and professional norms and understanding their differences could prove valuable when evaluating behaviors and motivations concerning integrating sustainable and resource-oriented sanitation. This reminder highlights the important difference between the public and the professional as we consider ROS and SS.

## 2. Methodology

The research process presented in this article was exploratory and based upon questions investigating nutrient recycling technologies and EcoSan within selected higher education programs across Europe. Specifically, two research questions were investigated using survey methods: one, whether there is a lack of knowledge about nutrient recycling technologies and EcoSan within the educational programs of future wastewater managers and sanitation engineers; and two, whether these technologies are important to the university educators responsible for wastewater education programs.

*2.1. Survey Methods*

2.1.1. Survey Participants

Given the intention of the survey to evaluate the extent to which ROS and SS are presently taught in Europe, an effort was made to reach as many academics involved in the coordination and teaching of civil and environmental engineering across Europe as possible. To evaluate which universities and programs to include in the survey, several online rankings were consulted. These included the global top 200 universities in civil and structural engineering of the QS-Ranking, the tool u-multirank, and the CHE university ranking for German universities, among others. The final list consisted of 151 universities and 294 study programs from the fields of civil engineering and environmental engineering with a focus on civil engineering; note, the exact title of these programs differed from university to university and country to country. Aside from programs in Civil Engineering (204), programs in Civil and Environmental Engineering (30), Civil and Structural Engineering or Civil Engineering and Construction (13), Environmental Engineering (20), and Civil, Structural, and Environmental Engineering (3) were addressed. In addition, a few rather specific programs in the field of water technology, infrastructural management, and sustainability science were selected (16). Finally, eight programs focused on general engineering. For each identified program, a faculty representative was sought; if, however, a direct faculty member could not be found, the study counselor or supervisor for the particular study program was addressed. One professional from each identified university was contacted (a total of 151).

2.1.2. Survey Questionnaire

The survey was concise, designed to focus on the specific questions of university programming, as well as to encourage participation (see Table 1). Survey questions were based on an interest and understanding of the scope of higher education educational programs for wastewater managers and sanitation engineers across Europe. A wastewater education faculty at Kristianstad University reviewed the survey questions prior to final inclusion.

**Table 1.** Nutrient recycling technologies and ecological sanitation program survey.

| Survey Questions |
|---|
| 1. Is ecological sanitation and recycling of phosphorus, nitrogen and potassium in form of struvite (ex.) part of your programme? |
|    a. IF SO, which percentage does this practice have in your lectures and courses approximately? |
|    b. IF SO, is this practice considered as a chance for _______ (the country the particular university is located in) or only discussed as a potential system in developing countries? |
|    c. IF NOT, why not? |
| 2. How important (very important, important, less important, unimportant) is this topic in your opinion? |
| 3. In which percentage do national, European and international students participate in your programme? |

Question 1 was designed to evaluate whether the concept of ecological sanitation, especially the recycling of nutrients, is discussed within wastewater treatment and sanitation courses of the particular program and university. Furthermore, participants were asked (a) to define the extent to which ecological sanitation specifically, and nutrient recycling in general, is taught and (b) to clarify the context. Here, the question was aiming to see whether the topics are shown in a national or international (developing countries) context. Question 2 asked for the personal opinion of each lecturer regarding the importance of the identified topic. The use of these questions together was done to be able to compare the results from Questions 1 and 2 regarding their congruency and to estimate the potential to introduce or further promote the topic of ecological sanitation and recycling of nutrients in an academic context. Finally, Question 3 asked for the share of national, European, and international students within the program. The question was designed to consider whether the knowledge stays

within the country it is taught or within the EU, or whether it may be "exported" to other countries after students graduate.

### 2.1.3. Survey Analysis

Frequencies and percentages were calculated using the survey data to support an understanding regarding the inclusion of nutrient recycling technologies and ecological sanitation within higher education programs across Europe. In a second step, the countries with most responses from programs and universities (five or more responding programs and more than two responding universities; see Figure 1) have been evaluated towards their current and potential future and the role of the three fixes to environmental behavior.

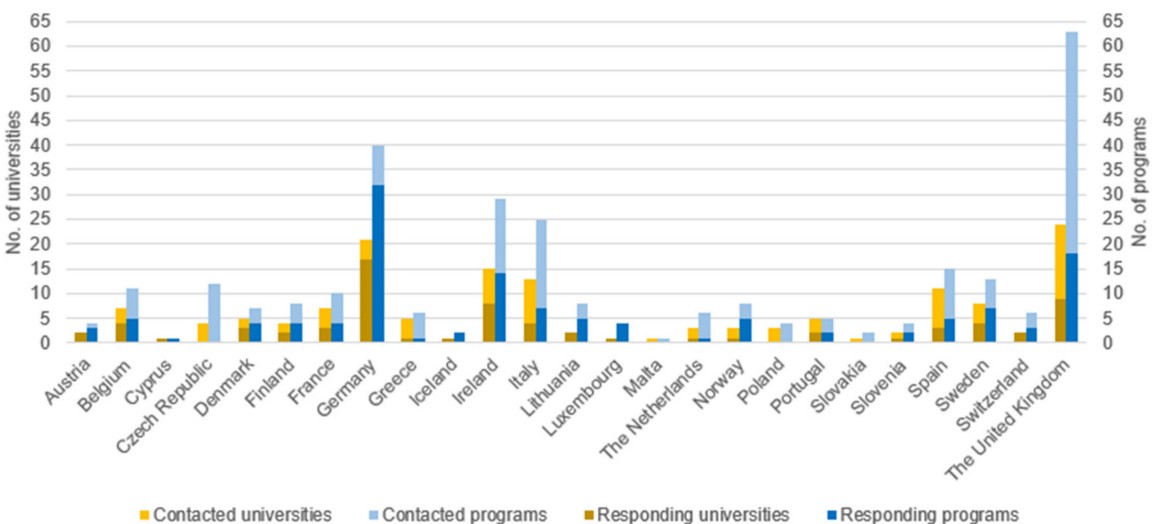

**Figure 1.** Responding universities and programs in relation to the number of contacted ones per country.

## 3. Results

### 3.1. General Evaluations

Figure 1 gives an overview of the contacted universities and programs per European country as well as the number of responses. On average, two programs per university were contacted. In Germany, for example, most universities had two programs—a Bachelor and a Master of Science in civil engineering. However, as the graph clearly shows, this ratio did not exist everywhere—the university of Luxembourg was contacted for four programs and the Norwegian University of Science and Technology for five. Correspondingly, other universities offered only one program, as can be seen in the cases of Cyprus and Malta.

Of the 151 contacted universities, 72 (48%) replied. These 72 responses related to 129 (44%) programs. Eighty-nine of these were in Civil Engineering (69%), 13 in Environmental Engineering (10%), 11 in Civil and Environmental Engineering (9%), eight in the fields of water technology, infrastructural management, and sustainability science (6%), three in Civil and Structural Engineering (2%), three in Engineering (2%), and two in Civil, Structural and Environmental Engineering (2%).

The responding 129 programs consisted of 62 bachelor programs, seven integrated master programs (a combination of Bachelor and Master in one program), 56 master programs, and four certificate programs. Almost 50% of all responding programs were located in only three of 25 countries—in Germany, Ireland, and the United Kingdom. No responses were received from the universities in the Czech Republic, Malta, Poland, and Slovakia.

### 3.1.1. Covering Ecological Sanitation and the Recycling of Nutrients in Education

A majority of 85 programs (66%) indicated that they teach about ecological sanitation and the recycling of nutrients. Forty-one programs do not teach about these topics, and three programs did not respond to the question. The following section provides an overview of the overall results.

### 3.1.2. The Extent of Coverage

A general evaluation of Question 1.1 is not as easy as one might have thought, due to a relatively high number of vague responses, as shown in Figure 2a. Forty-five of the 85 programs indicating instruction about the recycling of nutrients could not identify their percentage of coverage. Their answers included assessments like "very little," mentioning the minutes spent on the topic without relating it to the total hours taught or declaring the topic nonmandatory. The other 40 programs identified percentages from > 1% to roughly 50%. Among these 40 programs, 25 programs identified a coverage of 5% to 10%. The remaining 15 programs stated that their percentage coverage is lower than 5%, between 15% to 20% (six programs respectively), and at 25% (one program) and 50% (two programs). These percentages deviate from 15% to 40% of the median value and thus seem surprising. However, the 50% coverage takes place in courses at the master's level. These courses especially consider more specific topics like the one discussed here and present them with great detail. The 25% coverage takes place in a program with an emphasis on environmental issues. In comparison, most of the lectures mentioned by other programs are looking at sanitation in general and consider technological alternatives like the recycling of nutrients within these courses. The coverage is thus smaller in percentage because it is not the main subject but an addition or adaption to a more general topic.

### 3.1.3. The Context of Coverage

The evaluation of the second subquestion was more straightforward than the previous question. As Figure 2b shows, the responses can be divided into five categories: *no answer*, *both*, *in a national context*, *in the context of developing countries*, and *other*.

As one can evaluate, roughly one third (35%) of the responding programs teach about the topics of ecological sanitation and recycling of nutrients in both the national and developing country context. However, the technologies discussed might differentiate widely. For example, this evaluation corresponds with 42% of the programs in Germany. According to these programs, the focus for Germany and other developed countries is set more on high-tech recycling of nutrients from wastewaters. When looking at developing countries, however, the German faculties focus on modern sanitation concepts and the reuse of nutrient-rich water for irrigation as recommended with EcoSan.

A significant number of programs (28%) teach exclusively about nutrient recycling technologies in the context of their particular national situation. Here, the lecturers identified local or national regulations and guidelines as reasons for the national focus. Furthermore, companies that already apply technologies for wastewater recycling were mentioned as motivators for teaching about the subject. The portion of programs and universities that exclusively discuss the topics in the context of developing countries is rather small in comparison (5%). These programs stated that they consider the technology for the recycling of nutrients immature or too expensive. Finally, 7% of all programs reported that they consider the discussed technology in other contexts. One Italian university, for example, considered nutrient recycling as a potential for other developed countries such as Sweden. Other programs teach about the technologies and principles without identifying a specific context. Finally, 25% of all programs did not classify the context in which these topics are taught.

### 3.1.4. Reasons for Not Teaching about Nutrient Recycling

Roughly 32% of all respondent programs are not covering the topics of ecological sanitation and recycling of nutrients from wastewater. Out of these 41 programs, 17.1% did not give a reason for neglecting these topics (red column, Figure 2c). As can be seen in Figure 2c, the remaining 34 programs

identified ten different reasons. Color clustering has been chosen to evaluate the responses better. The yellow columns (26.8%) show reasons related to the particular syllabus—either not including sanitation courses in general or identifying other departments than civil engineering for the discussed topics. The blue columns (32.9%) combine programs that have sanitation courses but do not cover nutrient recycling and ecological sanitation. Their courses cover other environmental issues that are perceived to be more important, or they primarily teach about the technical status quo. Then, there are reasons related to academic resources like personnel, finances, or research focus (purple columns; 18.3%). Finally, the light green column (4.9%) shows two programs that plan to introduce relevant courses in the future

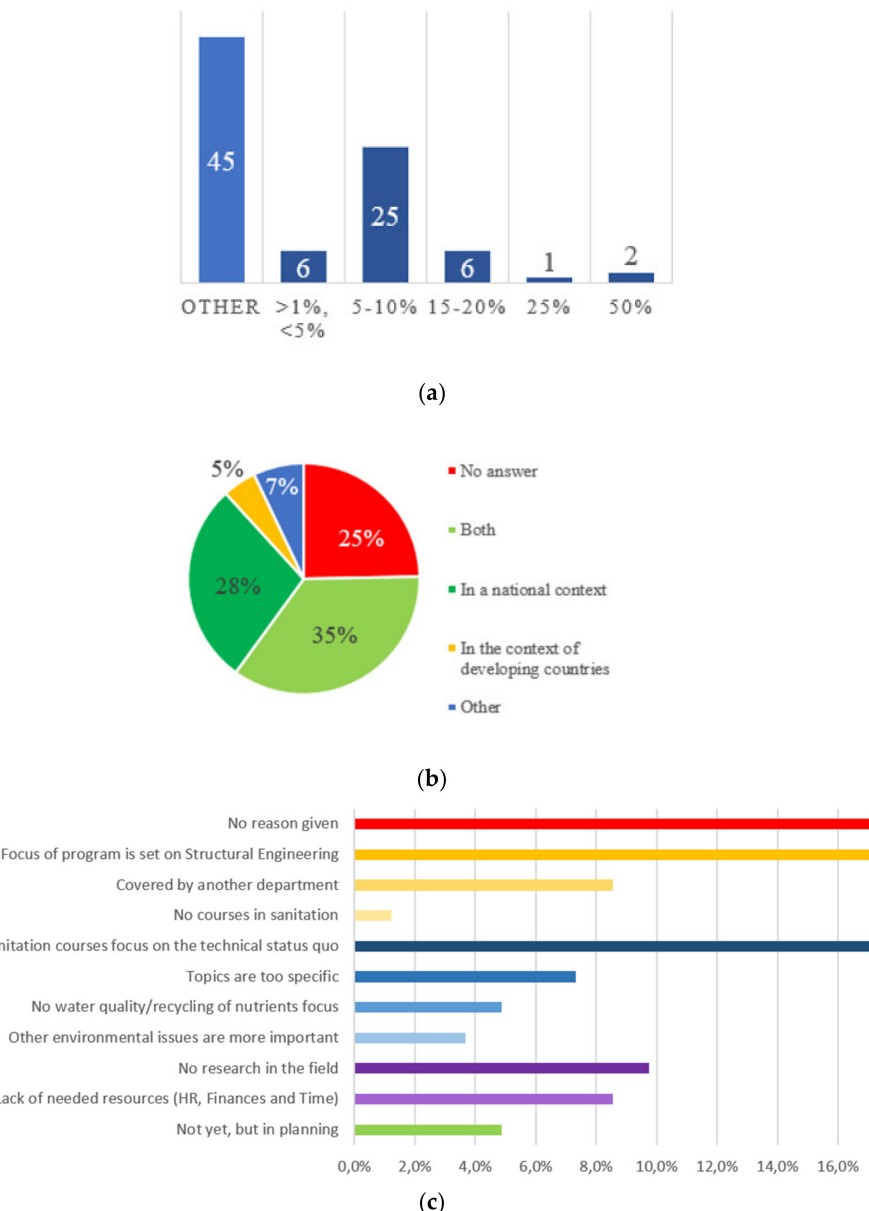

(a)

(b)

(c)

**Figure 2.** Responses to question 1 of the survey, see: (**a**) identifying with what percentage ecological sanitation and recycling of nutrients (P, N, & K) is taught (n = 85 programs); (**b**) showing the context it is taught in (n = 85 programs); (**c**) evaluating reasons for not teaching about the subjects (n = 41 programs).

### 3.1.5. The Importance of Nutrient Recycling and Ecological Sanitation

Question 2 was designed to understand how educators in the field of wastewater treatment view the importance of ecological sanitation and the recycling of nutrients. As Figure 3 shows, 74% of all responding lecturers identified the topic as important or very important. Five percent evaluated that the issue will increase in importance in the future.

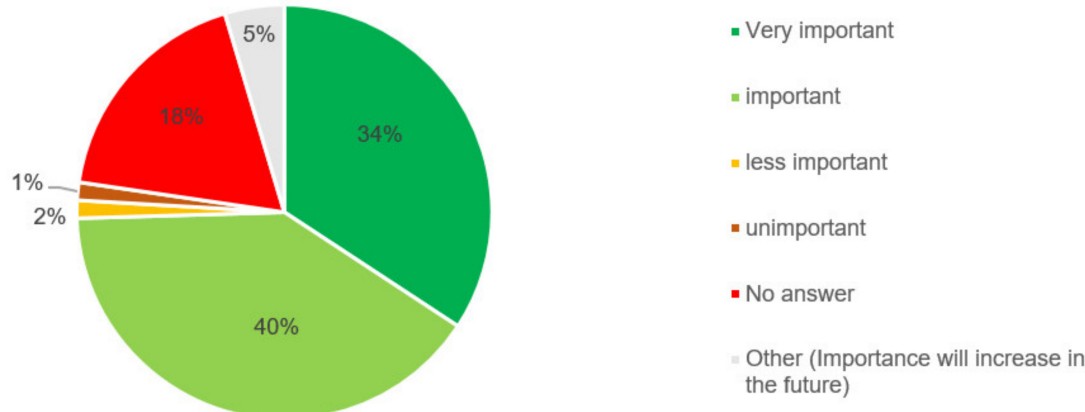

**Figure 3.** Percentage share of opinions about importance of nutrient recycling and ecological sanitation (answers to question 2 in relation to all respondent universities). (n = 72 universities).

### 3.1.6. The Percentage Origin of Students

Question 3 was intended to reveal the shares of national, European, and international students to evaluate whether there might be a national focus during the study programs and whether the learned knowledge might be exported after graduation. While in theory interesting, the evaluation of responses proved difficult. Over a third (36%) of the programs did not respond to this question or only gave vague answers to the effect that the courses are taught in their particular language and thus the fraction of national students might be high without identifying exact numbers. For the rest it can be said that a majority (54%) of programs had an above 50% share of national students. This majority consists of programs with a 90% to 100% (22%), a 70% to 90% (14%), a 60% to 70% (14%), and a 50% to 60% (4%) share of national students. The remaining 12 programs (10%) stated that their share of national students is about 10% to 50% (1), 33.33% (5), 25% (2), 10% (3) and 0% (1).

While the evaluation of national students was easily achieved, the evaluation of European and international students proved difficult. The responding programs often did not differentiate between European and international students and offered a cumulative percentage. Therefore, one could evaluate the number of international students by creating the difference of the above values. A differentiation between European and non-European international students, however, seems impossible and consequently so does a clear statement about "exported" knowledge.

Question 3 could not be fully evaluated due to missing information and a focus beyond the scope of this article, and will therefore not be discussed hereafter.

### 4. Discussion

The online survey showed that there is not a lack of knowledge about nutrient recycling technologies and ecological sanitation within the education of future wastewater managers and sanitation engineers. However, the degree to which these topics are taught differentiates from university to university and from country to country.

The majority of all responding programs consider nutrient recycling and ecological sanitation for both developed (national) and developing countries (35%) or solely in the national context (28%). Only 5% of the responding programs teach solely in the context of developing countries. According

to the responding faculties, the different contexts are connected with opposite technologies. While high-tech solutions and adaptions to the current WWTP system are commonly addressed in a national context, decentralized solutions such as ecological sanitation are considered for developing countries. Although there have been some examples of decentralized sanitation systems throughout Europe [18], this differentiation corresponds to the research literature [6,8,30]. Similarly, Takala [70] evaluated that while the research mostly discusses decentralized options and the separation of wastewater streams, practitioners in a developed country (Finland) perceive their centralized WWTP system as the place to enable sustainable development. Similar to the Phos4life project in Switzerland, the applied nutrient recycling technologies will thus have to meet the sustainability dimensions such as economic considerations and will differentiate from those promoted by EcoSan.

By looking at the results in Figure 2c, it can be evaluated that academic resources (purple columns) and changing priorities in the foci of sanitation courses (blue columns) are the main areas of action to further the education on nutrient recycling and ecological sanitation. Universities are said to play a vital role in societal transformations and should thus engage in topics of the SDGs, such as sustainable sanitation [35]. By allowing their faculties (and students) [37] to influence the education topics (bottom-up), while simultaneously creating support for specific topics (top-down) in the form of research funds or living laboratories [35,36], sanitation courses that are yet to mention nutrient recycling and sustainable sanitation might be motivated to do so in the future. Having living laboratories or green technology parks on campus enables technology transfer, which could impact the technical status quo and, subsequently, the education. In addition to these internal university strategies, there is a call for government policymakers to develop incentives that could encourage universities to strengthen their research and education towards topics of sustainable development [36]. A European fund for sustainability education could be an example of such an incentive. New EU and national policies might especially encourage change in programs with an above 50% share of national students and thus a relevant orientation.

A significant number of responses (see Section 2.1.2.) were achieved from the countries of Belgium, Denmark, Germany, Ireland, Italy, Spain, Sweden, and the United Kingdom (UK). The responses from Belgium, Denmark, and Germany showed that nutrient recycling and ecological sanitation are discussed in their courses. Except for two programs from Germany, their coverage is still rather small but could increase in the future if missing academic resources and finances are provided. In addition, organizational and societal developments are identified to affect future coverages of these topics. Here, the revision of the German sewage sludge regulation (AbfKlärV) in 2017 and setting a focus on water resource recovery facilities (WRRF) rather than WWTPs in Belgium were mentioned by responding faculties.

The responses from Ireland, Italy, Spain, and the UK predominantly pointed to the technological status quo when it comes to reasons not to address nutrient recycling and ecological sanitation. In terms of Italy and the UK, this result correlates with their distrust in science and technological innovation to enable environmental protection [71]. The responses of all four countries suggest that the topics of ecological sanitation and nutrient recycling will not gain more significance if the current technological situation does not change.

Finally, the results from Sweden were surprising. The coverage of nutrient recycling and ecological sanitation in education has been evaluated as relatively low (≤ 2%). This coverage contradicts Sweden's overall environmental awareness (67%) and trust in innovation (83%) [71,72] as well as its outside perception when it comes to these technologies. One Italian faculty, for example, stated: " . . . in the most developed European countries (including Sweden) it [nutrient recycling] is among the most promising themes for the foreseeable future and the one on which more investment in research is devoted." The reason for this discrepancy is unclear. One faculty stated that municipal WWTPs do not consider struvite in Sweden, while other faculties identified nutrient recycling as important for the sustainability of society rather than for education.

### 4.1. Considering a Combination of the Three Fixes to Environmental Behavior

The results of the survey showed that the technological status quo and structural problems were identified as the main impediments of a broader implementation of nutrient recycling in the form of EcoSan or SS. This leads to a discussion about the three fixes to environmental-friendly behavior. The fixes are presented as choices that are relatively independent of another, with the implication that in order to develop a specific environmental-friendly outcome, you choose one or the other [32].

The results of the survey seemingly suggest a linear approach where a combination of cognitive (academic education) and structural fixes (bottom-up & top-down approaches, both internal and external) set the stage for the technological fix (nutrient recycling, sustainable sanitation). Such a linear approach has been considered in the toilet revolution in China that requires incentive policies and transdisciplinary education [73]. It could be applicable in countries such as Denmark or Germany, where the main impediment to sustainable sanitation courses lies in academic resources and finances.

For countries such as Ireland and Italy, however, using a linear approach might not work as effectively. Here, the status quo inhibits all progress. Its resilient character [39] might require an interplay of the three fixes instead. Structural fixes that require technology developers (engineers) and educators (researchers) to work together could create synergies between the technological and cognitive fix. Seifert et al. [74] evaluated that WWTP-professionals are interested in cooperating with universities and research projects to foster innovation. Through measures such as incubators or living laboratories, education and technology innovation become feedback-driven and interdependent. This relationship may be able to affect the structural fix that might need to evolve and adapt to new functions and requirements. An incubator in the form of academic start-ups [75] or a network such as the European Sustainable Phosphorus Platform (ESPP) might set a focus on technology development first and later change to a management or operation focus. In addition, such participatory measures could help students to become less passive learners and more active change agents, triggering societal transitions [36]. It is our conviction that similar to the evaluations established by Krasney [52], all three fixes need to be part of a feedback-driven and interdependent (complex) system to enable environmental behavior. This approach is seemingly new, as no such system has been found in the current literature.

### 4.2. Understanding Attitudes and Emotions (Disgust)

Beyond suggestions that promoted a combination of the three fixes, some responses of the survey indicated educator attitudes as an important factor:

1. "The term ecological sanitation can mean many things to many different people. I have often associated it with not very good onsite sanitation systems, which require quite a bit of handling of fecal material, behavioral change, and risk of helminth transmission. If you think this an unduly negative view I strongly urge you to get some first-hand experience with someone else's fresh human excreta! However, the reuse of nutrients in wastewater and fecal wastes is, in principle, a good thing ... the range of workable technologies is limited. And this affects what we can teach." (English faculty)

2. "I think it is a very important topic which is why I give lectures on it to both undergraduate and post graduate students. I want the students to leave the College realizing that there is a big opportunity to recover valuable resources from wastewater." (Irish faculty)

3. "YES, because it is an important topic (I did my PhD and Habilitation about that topic!)" (German faculty)

An attitude is especially strong if rooted in personal experience and emotions (affect) [64]. Notions of personal responsibility and commitment are also important [7,27,32]. Response No. 1 above has been written from the basis of effect and potentially negative past experiences. The lecturer starts by pointing out what ecological sanitation means to him on a scientific level but continues with defending his opinion and recommending to get first-hand experience to understand what is asked

of him. He relativizes his statement by pointing out that the recycling of nutrients, in general, is a good idea and that the technological status quo strongly influences his lectures. The fact that he emphasizes others to get a personal experience with fecal matter reflects on the societal emotion of disgust [7,45–47]. Opposite to this statement, responses No. 2 and 3 both point to positive aspects of the topics. The second response emphasizes responsibility and commitment to educating students about the topic and its importance. This instructor's notion of responsibility seems to be part of his identity and role as a university professor. His students shall graduate with a clear picture of the opportunity for nutrient recycling from wastewater. He seems aware of his influence on the mindset of his students and feels obligated to act upon it. The lecturer of the third response emphasizes his personal experience with the topic as a reason to teach about it. Such a commitment will strengthen their role as educators and inspire others (students, the public) to live sustainably [35].

Promoting sustainable technologies such as nutrient recycling can trace back to positive attitudes such as respondents 2 and 3 showed and removal of thresholds such as academic resources (see Figure 2c). Seeing that Question 2 was mostly met by positive attitudes, a removal of thresholds could lead to an improved implementation of the discussed topics in education throughout Europe. However, the first respondent showed how deeply the emotion of disgust is still rooted in humankind. Disgust flags are very resilient [45], and thus a removal of thresholds would probably not change the degree of implementation where disgust influences opinions.

*4.3. Limitations*

During the survey, a significant number of participants did not understand the term "ecological sanitation." More than once, a definition was requested. Asking for ecological sanitation and the recycling of nutrients together seemingly increased the confusion. In addition, the term was met with criticism due to the energy and chemical input used to recycle wastewater for nutrient harvesting. Thus, the terms "sustainable sanitation" or "resource-oriented sanitation" might be more promising.

The evaluation of Question 1.1 was difficult. For future surveys, it is recommended to choose a different unit than a percentage. A possible measure would be to ask for a ratio (minutes taught to total hours of lectures). A future survey should also differentiate between Bachelor and Master programs, seeing that they differ in terms of study focus and available time—a low coverage in a master degree might still be more time spent than an equally low value within a bachelor program.

As some lecturers pointed out, other environmental topics might be prioritized rather than ROS and SS. A future survey should evaluate different aspects of the SDGs to evaluate the responses of Question 2 better. In addition, some of the responses received for Question 2 (see *4.3. Understanding attitudes and emotions (disgust)*) can emphasize the need to consider the social dimension when discussing sustainable sanitation, especially since such sanitation systems need to consider social acceptance among others to be considered sustainable. One question alone, however, is not enough to evaluate an attitude. A future survey would need to ask further and qualitative questions to assess involved attitudes and their vertical and horizontal structure [32].

## 5. Conclusions

- There is an apparent need to address the sustainability of the sanitation systems in Europe and around the world. In this context, the recycling and reusing of human wastewater under Target 6.3 of the SDGs must be promoted. While there is elaborate research about technological and ecological solutions, social factors have been investigated less vigorously. This article tried to evaluate the state of sanitation education in Europe when it comes to recycling practices. The research results show that knowledge about EcoSan and the recycling of nutrients within the higher education programs of civil engineering and related fields is not lacking, and that these topics are perceived as important.
- There are differences of considered technology when it comes to the geographical context of the education.

- The main impediment in education programs can be traced back to structural conditions and the technical status quo.
- In order to successfully implement the recycling of human wastewater, one needs to consider a feedback-driven combination of the three fixes to environmental problem-solving.
- Positive and negative attitudes may play an important role in terms of individual commitment and the emotion of disgust and require further research.

## 6. Outlook

> . . . *universities as a whole and university staff in particular, should try to take more advantage of the many opportunities SDGs provide to them, not only in respect to teaching and research but especially in respect of their outreach activities* . . . [35] *(p. 294)*

Universities and their faculties have a responsibility to operate in the interest of society to establish sustainable sanitation as a priority in the civil and environmental engineering education. Such efforts, specifically, a focus on Target 6.3 of SDG 6, provides a meaningful and tangible way to model how higher education institutions contribute toward the achievement of Agenda 2030 SDGs. Further opportunity for higher education institutions to align with Agenda 2030 in the field of wastewater management can be done through the creation of living laboratories or green technology parks [35,36]. Such examples highlight the interdependent nature of this problem-solving, i.e., state of the art technology (technological fix) supports the development of infrastructure (structural fix), and its application can be used to educate future wastewater engineers, and maybe even the broader public (cognitive fix).

In this context, one should evaluate how internal and external bottom-up and top-down approaches influence the education and technological innovation and vice versa. Germany and Switzerland could be useful case studies for such research, taking into account that they already promote circular economy approaches and revised regulations to improve their resource efficiency from wastewater in the future.

Attitudes and emotions towards the subject of sustainable sanitation need further evaluation. The emotion of disgust has been proven to be rather resilient [45] and may thus still widely influence our behavior towards concepts such as EcoSan; disgust may be a key moderator to explain the A-B relationship [64]. The emotion of disgust tends to strengthen the status quo, and thus there is a need for researching how inappropriate disgust flags could be repealed or managed. A greater understanding of the disgust emotion will also help broaden our efforts to educate beyond future wastewater engineers and include producers (farmers) and consumers (the public) in a better understanding of EcoSan and the use of recycled water and substances [41,76]. In addition, one needs to consider communication strategies and what terminology to use in order to positively frame the subject [41].

The research about nutrient recycling from wastewater is widely set on specific disciplines such as engineering in search of the appropriate technology or social sciences concerning perceptions and behavior. There is undoubtedly a need to look into transdisciplinary possibilities (integrated fixes, e.g., green technology parks) and the social dimension (attitudes & emotions), as well as further research about the future users of recycled nutrients. However, it is recommended that these research foci are integrated into a holistic research field that evaluates the linkages and dependencies between them.

**Author Contributions:** The research presented in this paper was a part of the first author's graduate studies at Kristianstad University, J.J.; The idea was conceptualized largely by the first author J.J. with support from the second author T.B.; J.J. was responsible for the methodology, formal analysis, investigation, and data curation. Both J.J. and T.B. were involved in the original draft preparation, writing, editing, and review. All authors have read and agreed to the published version of the manuscript.

**Funding:** This research received no external funding

**Acknowledgments:** We would like to thank graduate supervisor Lena B.-M. Vought, examiner Jean O. Lacoursière and all of the academic faculty that responded to the survey.

**Conflicts of Interest:** The authors declare no conflict of interest.

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
