# Peer review of "Ecological Sanitation and Sustainable Nutrient Recovery Education: Considering the Three Fixes for Environmental Problem-Solving"

_sustainability, doi:10.3390/su12093587_

Round 1
Reviewer 1 Report
It is an article with an interesting and relevant topic at present, but could be improved with the suggestions discussed below.
The objective, both in the summary and in the text, should be made clearer in the appropriate place, based on the research questions, and as it is focused on human and social factors.
In the introduction, it would also be convenient to relate it to the ODS of Circular Economy (indicated in the perspective section).
It is necessary to justify with the corresponding quote the statement on line 158 referring to the example of quality transport.
The role of action-oriented Environmental Education should also be argued (there is an abundant bibliography on this), and the role it can play in behaviors, for example in reducing water consumption, leading to less wastewater treatment .
As for the methodology, do not put materials as a title, put it as a subsection. In the methodology it must be made explicit and justified because it is appropriate to use surveys, which criteria were followed to select the questions.
Why is what is said in the summary of the results repeated at the beginning of the discussion?.
Obviously the limitations of the study are significant, but it is appreciated that they are considered by the authors, what is expected is that they delve into it in future research.
It would not hurt to put a section of conclusions, where the most relevant conclusions of the work are highlighted, depending on the proposed objective, since some of them are exposed in the perspective section.
Author Response
Manuscript ID: sustainability-773785
April 20, 2020
Julian Junghanns
Thomas Beery
Review 1
It is an article with an interesting and relevant topic at present, but could be improved with the suggestions discussed below.
The objective, both in the summary and in the text, should be made clearer in the appropriate place, based on the research questions, and as it is focused on human and social factors.
In the introduction, it would also be convenient to relate it to the ODS of Circular Economy (indicated in the perspective section).
We are not sure what you mean when you refer to the “ODS” of Circular Economy. A Google Search has presented us with two options. Either you reference Operational Data Stores and thus the relationship of the Circular Economy to the digitalization, or ODS is the Spanish equivalent of the SDGs (Objetivos de Desarrollo Sostenible, ODS) and you are recommending to reference the importance of CE in the context of the SDGs. For the latter option, we have included additional information on the importance of CE for achieving the SDGs and its targets such as the recycling of wastewater under 6.3. For this new information, check line 66-69.
It is necessary to justify with the corresponding quote the statement on line 158 referring to the example of quality transport.
In connection to the second review, we have substituted this statement with one closer related to the water management field and provided a reference. This can be found at line 178-180.
The role of action-oriented Environmental Education should also be argued (there is an abundant bibliography on this), and the role it can play in behaviors, for example in reducing water consumption, leading to less wastewater treatment.
This is a good recommendation and we have added a whole section for it at line 154-169, referencing the development of environmental education from a merely knowledge based concept to one that considers a multitude of factors to enable behavioral change.
As for the methodology, do not put materials as a title, put it as a subsection. In the methodology it must be made explicit and justified because it is appropriate to use surveys, which criteria were followed to select the questions.
We have deleted “Materials” (see line 240) from the title and added “Survey questions were based on an interest and understanding of the scope of higher education educational programs for wastewater managers and sanitation engineers across Europe. A wastewater education faculty at Kristianstad University reviewed the survey questions prior to final inclusion.” at line 269-272 to explain and justify the chosen survey question.
Why is what is said in the summary of the results repeated at the beginning of the discussion?.
In relation to Reviewer 2, we have deleted the section “Summary of the Survey Results”, and thus removed this repetition. We felt that it needs to stay in the discussion section because it offers a good starting point for this section. These changes can be seen at line 409-418.
Obviously the limitations of the study are significant, but it is appreciated that they are considered by the authors, what is expected is that they delve into it in future research.
It would not hurt to put a section of conclusions, where the most relevant conclusions of the work are highlighted, depending on the proposed objective, since some of them are exposed in the perspective section.
We have added a conclusion between the discussion and the outlook that summarizes the main results with the help of bullet points at line 567-585.
Reviewer 2 Report
The manuscript “Ecological sanitation and sustainable nutrient recovery education: considering the three fixes for environmental problem-solving” reports an interesting survey results based on education curriculum containing ecological sanitation and nutrient recycling European-wide survey with universities that offer civil and environmental engineering programs. Though, online survey has limitation to explore a topic in a holistic way.
Line 71-72, “Urine offers excellent fertilizer raw material due to its high fraction of nutrients.” Urine offers high fraction of Urea, not any many ‘nutrients’.
Line 156-159, This statement is not relevant to the context of study.
Line 256-260, Although the authors mention that query 3 is not relevant to the present study, however, a short overview of results could be informative and would elude incomplete sense of report.
Line 314-315, and 326-327, “Overall, the topics of ecological sanitation and recycling of nutrients are taught in both the national and developing country context 315 (35%).” And “At 5%, the portion of programs and universities that teach the topics solely in the context of 326 developing countries is rather small.” These statement seem contradictory.
Line 358-367, Section 3.2. Summary of the Survey Results is already covered in discussion part. This section is suggested to be removed.
Author Response
Manuscript ID: sustainability-773785
April 20, 2020
Julian Junghanns
Thomas Beery
Review 2
The manuscript “Ecological sanitation and sustainable nutrient recovery education: considering the three fixes for environmental problem-solving” reports an interesting survey results based on education curriculum containing ecological sanitation and nutrient recycling European-wide survey with universities that offer civil and environmental engineering programs. Though, online survey has limitation to explore a topic in a holistic way.
Line 71-72, “Urine offers excellent fertilizer raw material due to its high fraction of nutrients.” Urine offers high fraction of Urea, not any many ‘nutrients’.
I understand the critique. The idea here was to address a quality rather than a quantity issue. Obviously, urine consists mostly of water and urea in terms of quantity. Our focus, however, has been set on the quality part and the percentage share of NPK in urine in comparison to feces. To make this more understandable, I have deleted this sentence and substituted it with 2.5 sentences that describe the quality aspect with more detail. For this have a look in the new submission file at line 75-78.
Line 156-159, This statement is not relevant to the context of study.
You are correct. In accordance to Reviewer 1 who has also criticized this part, I have changed the example given to one of water management significance. For this have a look at line 178-180.
Line 256-260, Although the authors mention that query 3 is not relevant to the present study, however, a short overview of results could be informative and would elude incomplete sense of report.
As you can see on page 7, line 287/288, we have deleted the sentence “This inquiry, while interesting, is not entirely relevant to this study and is not included in the analysis.”. To your critique we have added a short overview of the results of query 3 under 3.1.6 at line 390-408.
Line 314-315, and 326-327, “Overall, the topics of ecological sanitation and recycling of nutrients are taught in both the national and developing country context 315 (35%).” And “At 5%, the portion of programs and universities that teach the topics solely in the context of 326 developing countries is rather small.” These statement seem contradictory.
They are not contradictory but a description of figure 2 b). I changed the wording a bit to hopefully make it less confusing. 35% of the responding programs teach about the topics in both contexts – they address both technical applications for developing countries and industrial countries such as their own in their lectures. 5% of the responding programs do only consider the developing country context during their lectures. In comparison to programs that teach about both contexts (35%) and those that teach exclusively about their own national context (28%), these 5% seem rather small. The adapted sentences can be found at line 344-346 and line 357-358.
Line 358-367, Section 3.2. Summary of the Survey Results is already covered in discussion part. This section is suggested to be removed.
We agree with your evaluation and have deleted the section 3.2, as you can see at line 409-418.
Reviewer 3 Report
This is a readable paper. My only suggestion is it would be better if the authors could modify Figure 2c. The colors made readers confused.
Author Response
Manuscript ID: sustainability-773785
April 20, 2020
Julian Junghanns
Thomas Beery
Review 3
This is a readable paper. My only suggestion is it would be better if the authors could modify Figure 2c. The colors made readers confused.
We have made adjustments to address this concern as you can see in the new uploaded submission at line 378. I have rearranged the results of Figure 2c to ensure there is still a color clustering. This clustering, however, is now achieved by color gradations, so that every reason got its own color. I hope this improves the figure to the point where it is easily understandable.
Round 2
Reviewer 1 Report
I congratulate the autores for their concise and adequate response to the sugestión mad, for my part can be aceppted for publicación.
Reviewer 2 Report
This revised manuscript provides satisfactory response to reviewer's queries, and recommended for acceptance.